# Unstructured satellite survey detects up to 20% of archaeological sites in coastal valleys of southern Peru

Thomas J. Snyder[1]*, Randall Haas[2]

1 Department of Anthropology, Evolutionary Wing, University of California, Davis, Davis, CA, United States of America, 2 Department of Anthropology, University of Wyoming, Laramie, WY, United States of America

* tjsnyder@ucdavis.edu

## Abstract

Satellite survey is widely used for archaeological site discovery, but the efficacy of the method has received little systematic investigation. In this analysis, twelve study participants of different experience levels performed an unstructured remote survey of 197 km$^2$ in the Sama and Moquegua valleys of south central Peru where previous pedestrian surveys recorded 546 archaeological sites. Results indicate an average site discovery rate of 9.3% (0–18%, 95% range). The most experienced participants detect up to 20% (17–22%) of known archaeological sites. These detection rates can be used to derive reliable site frequency estimates on the Andean coast, which can be used in planning and budgeting for field efforts and estimating demographic patterns at large spatial scales that are difficult to achieve through pedestrian survey. More generally, this analysis offers a method for deriving correction terms specific to other parts of the world. Additionally, the results can serve as a baseline for evaluating the effectiveness of emerging artificial intelligence routines for archaeological site detection.

**Data Availability Statement:** All relevant data are within the manuscript and its Supporting Information files.

**Funding:** The authors received no specific funding for this work.

## Introduction

Ever since the use of plane-based aerial photography in Gordon Willey's landmark Virú Valley Project, archaeologists have increasingly relied on remote sensing technologies to answer archaeological questions [1]. Satellite imagery in particular has been used investigate a diverse range of research foci ranging from anthropogenic landscape modification to exchange network analyses to the monitoring of the looting of archaeological sites over time [2–4]. In recent decades the technological sophistication employed in remote survey has dramatically increased, changing the ways in which we are able to discover and examine archaeological phenomena. Whereas remote sensing technologies were originally limited to the human observation of photographs, advances in satellite and imaging technologies, such as LiDAR, have facilitated groundbreaking discoveries, revealing the extent of Maya cities in Yucatan [5] and the existence of large villages in the Amazon [6]. Today, remote sensing studies incorporate a wide range of technologies, methods, and theories [7–10].

**Competing interests:** The authors have declared that no competing interests exist.

Despite such rapid technological and methodological advances, one of most basic and useful aspects of remote sensing technologies, particularly in places with little or no forest cover, remains the visual detection of archaeological sites. Traditional pedestrian survey methods and advanced remote sensing methods can be costly, both in terms of time and money. In contrast, high-resolution satellite photography has become virtually free in recent years, requiring only a computer and internet access. This ease of accessibility of satellite and aerial photography has led to its widespread adoption by archaeologists. Free and publicly available software such as Google Earth and Bing Maps provide satellite images with resolutions down to 61 centimeters [11, 12]. Similarly, within Middle Eastern contexts, the declassification of high resolution satellite imagery from the Cold War, as part of the CORONA Atlas Project, has provided archaeologists with a wealth of high-quality satellite imagery for research use [13, 14].

Another utility of satellite imagery is the ability to monitor looting and the destruction of archaeological sites over time [12, 15, 16]. While it is often not possible, practical, or safe, to monitor looting on the ground, satellite imagery allows archaeologists to identify incidents and quantify rates of site loss by comparing satellite images taken at different times [4, 11, 12, 15, 17]. This has been especially useful in the Middle East, where CORONA imagery has allowed for extremely precise analysis of the rate at which sites are being looted or being destroyed by development and agricultural practices [18]. Legacy images similarly allow archaeologists to track natural taphonomic processes [14, 19–22].

The utility of satellite imagery tends to be greatest where societies constructed highly visible architecture and in desert regions where vegetation is minimal [10, 22, 23]. Satellite surveys in the desert environments of Egypt and Peru demonstrate how the method can complement more traditional pedestrian survey by helping to prioritize areas based on environmental factors that might have affected land-use patterns or may affect archaeological accessibility [10, 24]. While satellite survey will certainly fail to observe some proportion of archaeological sites–especially small artifact scatters–it makes up for this shortcoming by allowing coverage of large areas at little to no cost, thus increasing detection rates for large sites.

Scholars have also begun to extend the utility of satellite imagery with machine learning techniques, training computers to automate the process of site discovery [25–27]. The approach promises efficient identification of sites, particularly in regions where it may be costly or impossible to survey using conventional methods [17, 24, 28]. However, these types of automated satellite surveys have a number of limitations, and previous research demonstrates that the most effective machine learning programs are still outperformed by teams of experts manually combing through large volumes of high resolution satellite imagery [28, 29].

For example, one automated satellite survey by Menze and Ur (2012) reports a success rate of 70–90% for identifying known archaeological sites, but also erroneously identifies a high volume of modern archaeological and geological features–with nearly 40% of all identified features being false positives [20]. Furthermore, Menze and Ur rely on multispectral imagery in addition to visible photography, rather than on just satellite photography. Casana (2014) convincingly demonstrates that the most effective form of satellite survey is what they term the 'brute force' approach–a team of experts painstaking reviewing high volumes of satellite imagery, though their study did quantify detection rates [29].

Crowdsourcing approaches for the remote detection of archaeological sites have also been employed, but like automated searches, face a number of issues [10, 30]. Foremost among these is that the general audiences employed in crowdsourcing approaches lack the anthropological, archaeological, or regional training needed to search successfully for archaeological sites. To both the lay person and automated computer algorithms, it can be difficult or impossible to distinguish between archaeological and modern features [22, 28, 30]. Crowdsourcing and automated approaches may make up for these shortcomings with the increase in coverage

as computer programs and large groups of people are collectively capable of surveying satellite imagery much faster than small groups of experts [30, 31]. Modern development can further hamper satellite survey, potentially obscuring archaeological sites and rendering them invisible to satellite photographs [32].

While 'brute force' and automated approaches offer distinct advantages for discovering archaeological sites in high resolution satellite imagery, they are also time and resource intensive–often beyond the scope of research budgets [29, 33, 34]. For most researchers, informal unstructured survey approaches utilizing freely available satellite imagery is more accessible and therefore most likely to be used prior to conducting more formal pedestrian survey. Archaeologists routinely use unstructured (i.e., informal) satellite survey as a matter of convenience. This study aims to examine the effectiveness of such unstructured satellite surveys in order to quantify their utility. By counting the proportion of archaeologically known sites that satellite surveyors can identify, we can speak to Stewart et al.'s (2020) call for an analysis on the biases and factors impacting crowdsourcing approaches to remote survey [35]. While previous studies have quantitatively demonstrated the utility of highly structured, systematic satellite surveys performed by teams of experts, the quantitative efficacy of unstructured satellite survey—the most pervasive form of satellite survey—remains unknown as does the extent to which experience affects survey efficacy [13].

Quantitative specification of unstructured satellite survey efficacy offers a formal approach to estimating actual site frequencies without the time and effort needed to organize pedestrian or 'brute force' approaches. At most, such unstructured approaches may make satellite survey useful for answering substantive questions about past human demographics and land-use patterns. At least, they provide site frequency estimates in advance of pedestrian survey or structured satellite survey, which may be useful for time and financial budgeting. Unstructured surveys may be especially useful for junior scholars without the time or finances to dedicate to large-scale systematic surveys or the connections required to organize teams of experts.

This study also examines the specific factors that potentially affect success in unstructured satellite survey. We examine the impact of the education and site type on site-discovery rates. More experienced surveyors are more likely to recognize archaeological sites and distinguish them from modern features. Archaeological site age also may affect probability of discovery, as it stands to reason that older sites will be less frequent and visible than newer sites [36]. Specification of these effects allows for greater precision in estimating actual site frequencies and will help researchers select survey teams appropriate to their analytical goals prior to engaging in field research.

While there are no pre-existing models to suggest the proportion of archaeological sites that can be found through unstructured satellite survey, we expect the proportion to be low given that the resolution of freely available satellite imagery may be too low to detect small sites, which tend to be the most frequent. For example, archaeological sites frequently consist of artifact scatters, such as lithic or ceramic scatters, which would be visible to pedestrian surveyors but invisible to satellite surveyors. We further hypothesize that greater archaeological experience will predict greater success at finding archaeological sites through satellite survey.

## Materials and methods

To investigate our hypotheses, we analyzed unstructured satellite survey results from study participants surveying for archaeological sites in a desert region in south central Peru. To assess the effects of experience on site detection rates, we recorded participants' degrees of archaeological training, experience with geographic information systems (GIS), research experience in the region, and pedestrian survey. All participants used the same imagery

from Bing Maps and record their findings using QGIS software [37]. The results of the unstructured surveys were analyzed using a series of generalized linear models within the R statistical programming language [38]. We describe each component of the analysis here, including the study region, survey method, archaeological controls, and method of statistical assessment.

## Study region

The study region for this project consisted of two coastal valleys in Southern Peru–the Osmore drainage and the Sama Valley, located at approximately 17.8° south latitude, 70.8° west longitude. This region was chosen for three reasons. First, the arid Atacama Desert environment offers exceptional preservation with sites occurring on the surface and known to be detectable in satellite images. Second, the history of reported archaeological investigations in these regions provides the ground control needed to validate satellite survey results. Finally, the lack of dense surface vegetation provides ideal conditions for remote detection.

Despite the extreme aridity of southern coastal Peru, the Osmore drainage has been host to a number of societies spanning over 12,000 years [39, 40]. These include but are not limited to the Huaracane, Tiwanaku, Wari, and Estuquiña in the upper Moquegua valley, and Chiribaya in the lower Ilo valley [41–44]. All of these more recent archaeological cultures produced ceremonial, domestic, and funerary architecture visible on the surface and in freely available satellite imagery. In the pedestrian surveys used as the control for this study, agricultural features were not recorded. This is likely because dating such features in the Andes can be problematic, as agricultural features are frequently re-used over time and by contemporary agro-pastoral populations, thus making the distinction between modern and archaeological features difficult.

## Satellite survey

Study participants with varying levels of archaeological experience are recruited to perform the satellite survey of the archaeological regions described above. In order to assess the effects of experience on site detection, participants are asked about their regional specialty, degree of archaeological training (undergraduate student, graduate student, Ph.D.), experience with GIS, research experience in the region, and whether or not they had ever participated in a pedestrian survey. For the purposes of this study, educational experience is tiered into undergraduate student, graduate student, or Ph.D., and other educational experience is treated as binary. Individuals either had the specified experience, or they did not. After answering these questions, participants then proceed to complete the satellite survey.

Our volunteer participants were taught to mark archaeological sites using QGIS and asked to note the amount of time that passes between locating consecutive sites. Participants were also provided with sample imagery showing the locations of 6 sites in a nearby region to familiarize them with the software, the process of marking features, and basic site identification. Participants were not instructed to follow a specific search method. Similarly, participants were not given a time limit for completing the satellite survey.

By conducting an unstructured survey with minimal guidelines for what qualifies as an archaeological site, how to search, or the scale at which to search, we evaluate the types of unstructured satellite survey performed by scholars with a range of experience. This approach allows archaeologists to not only understand the efficacy of unstructured satellite survey but also the extent to which general experience affects survey outcomes and training can mitigate those effects.

## Archaeological ground control

The satellite survey consists of three regions for which pedestrian survey has been completed in the past–the Lower Ilo Valley region survey area reported by Owen [45], the Moquegua Valley survey area reported by Goldstein (2005), and the Sama Valley reported by Baitzel and Rivera Infante [46] (Table 1 and Fig 1). Archaeological sites from these surveys serve as ground control for this analysis. The Lower Ilo Valley Survey Area covers 30 km$^2$, with a narrow band of vegetation on either side of the Ilo river and a middling density of archaeological sites. Survey spacing was conducted at 30 m transects. Chiribaya sites feature most prominently. The Goldstein Survey area encompasses the Moquegua valley, and is the largest of the three survey areas, with large architectural features at a high density. Tiwanaku sites, such as Omo M10, feature most prominently. Finally, the Baitzel-Rivera Infante Survey Area is the smallest of the three, covering 7 km$^2$, but with the highest density of archaeological sites. Survey spacing was conducted at 8 m transects with near 100% visibility due to the lack of ground cover. Unfortunately, survey methods are not reported for the Goldstein (2005) survey, but we assume here that the survey methods and results are comparable.

All satellite surveys are conducted using Bing satellite imagery viewed in QGIS [37]. Unfortunately, Microsoft does not report the date or resolution of satellite imagery. While this dearth of information precludes analysis of the effects of image date or quality, it reflects the reality of how archaeologists use satellite imagery in unstructured survey. The satellite survey area is constrained to the survey boundaries of the Owen, Goldstein, and Baitzel-Rivera Infante pedestrian surveys as digitized from their published maps using QGIS's map georeferencing tool (S2 File).

Site locations and types are digitized from the maps. The known-site database incorporates 546 sites from the three survey areas (S3 File; see Table 1). All geographic data are referenced to the World Geodetic System (1984) and projected to the Universal Transverse Mercator System, zone 19S. The ground surveys chosen for comparison with satellite survey in this study do not represent the full archaeological record of the study areas, nor do they comprise the complete history of archaeological survey in each region [47–49]. However, the purpose of this study is to compare the results of satellite survey and specific ground survey events, rather than to compare satellite survey with the history of archaeological survey in a region. Additionally, sites within the study region vary greatly by size, type, composition, and preservation. While the effect of these parameters on the success of satellite survey is an interesting question worthy of research, it is outside the scope of the presented research.

The frequency of known-sites discovered by each participant is used to derive discovery rates. Determining whether a given site was discovered entailed several analytical steps. The map georeferencing process imparts some spatial error in site locations on top of any error inherent in the survey maps themselves. Furthermore, the satellite images, from which site locations are derived in the survey, impart some spatial error. To assess the combined spatial error between the satellite images and georeferenced survey maps, we measure the offset distance between the locations of a sample of digitized archaeological sites and their

**Table 1. Summary of pedestrian survey data used as ground control in this study.**

| Project | Area (km$^2$) | Site Count | Citation |
|---|---|---|---|
| Moquegua Archaeological Survey Project | 160 | 431 | Goldstein 2005 |
| Sama Archaeological Survey | 7 | 47 | Baitzel and Rivera Infante 2019 |
| Proyecto Colonias Costeras de Tiwanaku (Ilo Valley) | 30 | 68 | Owen 1992 |
| Total | 197 | 546 | |

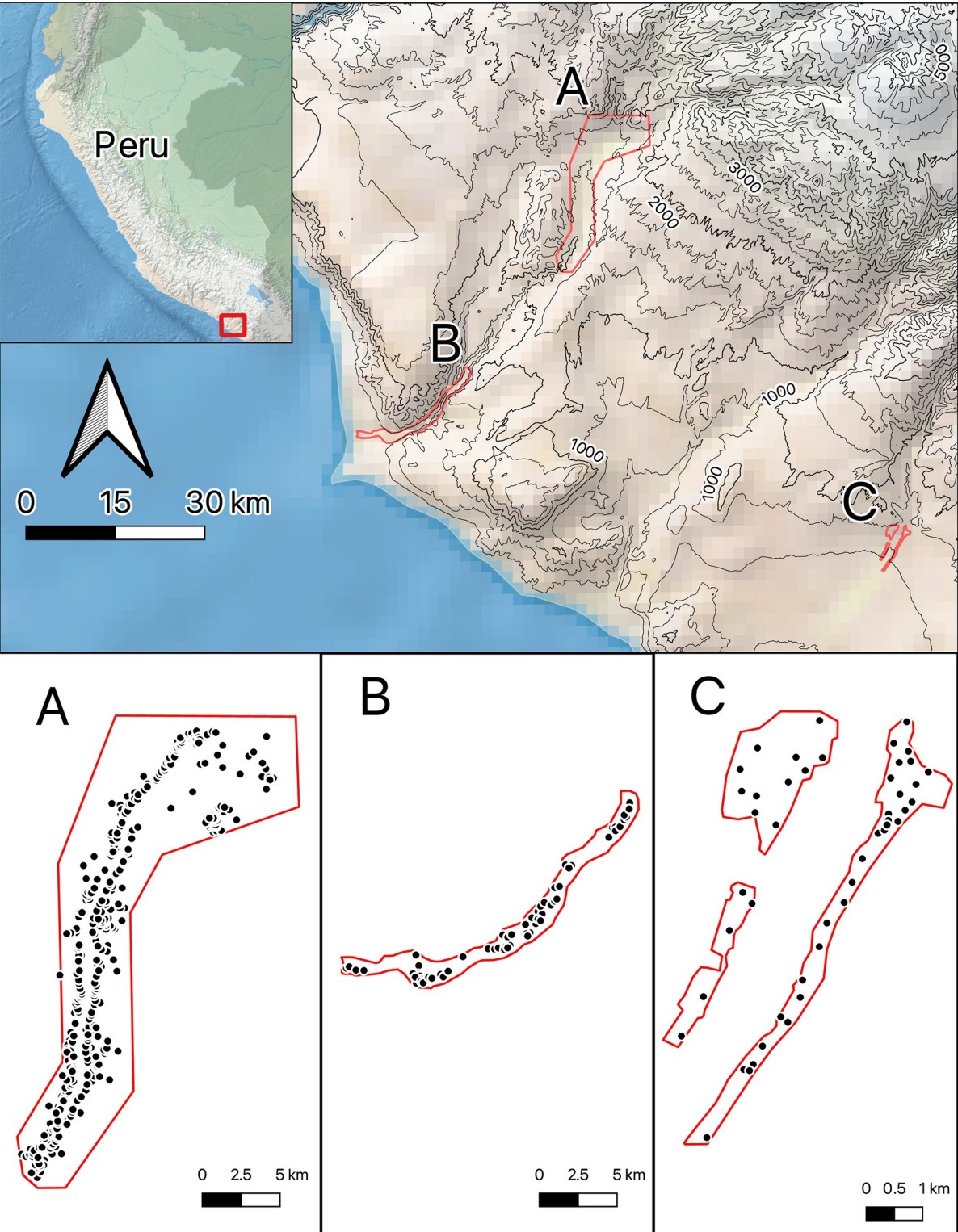

**Fig 1. Survey regions and archaeological sites used in this study.** Map depicts study area in the coastal south-central Andes of Peru, and archaeological survey outlines as follows—A: Middle Moquegua survey by Goldstein (2005), B: Tacna survey by Baitzel and Rivera Infante (2019), and C: Lower Ilo survey by Owen (1992). Made with Natural Earth and elevation data from the United States Geological Survey GMTED.

corresponding positions on Bing Maps satellite imagery. We do this for 30 random sites that are clearly visible in the satellite imagery. A lognormal distribution model is then fit to these error measurements and used to derive a 95% quantile value, which is used to define the error threshold. Our error analysis suggests a typical error of less than 118 m. In other words, the analysis shows that the geographic error between the georeferenced archaeological sites and their corresponding positions in satellite imagery is less than 118 m 95% of the time. We use this value as a threshold for establishing whether a surveyor correctly identifies an archaeological site or marks a false positive. If a satellite surveyor's mark falls within 118 m of an archaeological site identified by previous pedestrian survey, it is marked as found. If a marked site does not have a known site within 118 m, it is marked as a false positive.

Not only does our study evaluate a satellite surveyor's ability to detect known sites but also their propensity to identify false positives. Our definition of false positive assumes that prior pedestrian surveys will necessarily be more reliable than satellite surveys. Although there are situations where satellite survey can be more effective than pedestrian survey at site discovery, within our study region this is unlikely to be the case given the intensity of archaeological survey in a region with high surface visibility.

For each archaeologically known site, successful detection is recorded as 1 and failure as 0 for each survey participant. An individual's false positive rate is then calculated as the proportion of false positives identified among the total count of sites identified. In order to model the distribution of detection and false-positive rates among all participants, beta distribution models are fit to participant rates using maximum likelihood estimation. Kolmogorov-Smirnov (KS) tests are used to assess goodness-of-fit between model and data. Models that produce KS $p$ values greater than 0.1 are considered acceptable approximations of discovery and false-positive rates and thus used for rate estimation.

Although the Beta distribution is the appropriate statistical model for aggregate rate data, the alpha and beta shape parameters lack the intuitive appeal of mean values associated with normal distributions. In other words, we would like to know the average discovery and false positive rates among our participants. We therefore convert the alpha and beta parameter values to a mean, which identifies the most-likely rate in the population, using the following equation [50]:

$$mean = \frac{\alpha}{\alpha + \beta}$$

We furthermore characterize discovery rate variance as a 95% quantile range based on modeled beta distributions. We choose this variance metric to allow for asymmetries around the mean and thus avoid complications with low values, which potentially create meaningless negative site discovery estimates. These 95% ranges are expressed parenthetically following the calculated mean values in our presented results.

In order to assess the effects of experience variables—education, Andean archaeology, pedestrian survey, and GIS—we use generalized linear mixture modelling (GLMM) using the Satterthwaite approximation applied to restricted maximum likelihood fitted models as implemented with the lme4 and lmerTest packages in R statistical computing environment [38, 51, 52]. As the site-discovery data are binary, success is modeled using the binomial family of distributions. The model specifies archaeological site and surveyor as random effects. GLMM is also used to determine the effect of the additive variables on false positive rates but using the Poisson family of distributions because the outcome is a ratio as opposed to binary in the site detection analysis. Again, archaeological site and surveyor are specified as random effects.

Two additional binomial GLMMs are also run—the first to determine the impact that archaeological site age has on discovery rates. The latest known occupation for an

**Table 2. Satellite survey summary data, proportions (%) presented in parentheses.**

| Participant | Education | Andes | GIS | Survey | Sites found | False positives |
|---|---|---|---|---|---|---|
| 1 | undergrad | yes | no | yes | 25 (0.04) | 6 (0.19) |
| 2 | undergrad | no | no | no | 6 (0.01) | 1 (0.04) |
| 3 | undergrad | no | yes | no | 24 (0.04) | 2 (0.07) |
| 4 | undergrad | yes | no | yes | 46 (0.08) | 18 (0.28) |
| 5 | grad | no | no | yes | 19 (0.03) | 14 (0.42) |
| 6 | grad | yes | no | no | 92 (0.16) | 66 (0.42) |
| 7 | grad | yes | yes | no | 72 (0.13) | 1 (0.01) |
| 8 | grad | yes | no | no | 20 (0.03) | 6 (0.23) |
| 9 | phd | yes | no | yes | 100 (0.18) | 31 (0.24) |
| 10 | phd | yes | yes | yes | 107 (0.20) | 29 (0.21) |
| 11 | phd | no | yes | yes | 33 (0.06) | 3 (0.08) |
| 12 | phd | yes | no | yes | 65 (0.12) | 9 (0.12) |

archaeological site is a fixed effect in this model. The second is to determine the impact of archaeological site type on discovery rates, in which case, site type is specified as the fixed effect. All statistical analyses are performed in R statistical computing environment [38].

## Results

Study recruitment resulted in 12 participants including four undergraduate students, four graduate students, and four Ph.D. scholars from the University of California, Davis–with two outside participants (Table 2). All participants are archaeologists or participated in some form of archaeological course work and research. Eight participants identified themselves as specialists in Andean archaeology, seven had experience with pedestrian survey, and four had experience working with GIS in a research capacity. The participants identified a total of 824 archaeological sites, including false positives (S4 File). This count includes the same site multiple times if found by different participants. These data provide the basis for estimating site-detection rates, false-positive rates, and the effects of surveyor experience on those rates. However, the relatively small participant pool raises concern about the ability to accurately characterize effect sizes. A recent simulation study shows that although GLMM is relatively robust terms of detecting effects, the effect of sample size on accurate quantification of effect size remains poorly understood [53]. We therefore proceed with confidence in interpreting the model's ability to detect relationships but with caution in terms of the model's ability to quantify effect sizes.

### Site discovery rates

Site discovery rates ranged from 1–20% among participants (see Table 2). A best-fit beta model produced shape parameters of $\alpha = 1.77$ and $\beta = 17.39$ (Fig 2). A KS goodness-of-fit test shows the beta model to offer a plausible model for the data (KS D = 0.15; $p = 0.90$). From the modeled Beta distribution, we derive a mean discovery rate of 9.3% (1–18%).

Our binomial GLMM results demonstrate that the following variables significantly ($p < 0.1$) affect site detection rates: educational experience, experience in Andean archaeology, experience working with GIS, and time spent surveying (Table 3). Pedestrian survey experience is not observed to produce a significant effect. The strongest effect is found in Andean experience with training in Andean archaeology producing a 335% increase in discovery rate from a median site discovery rate of 4% to 13%.

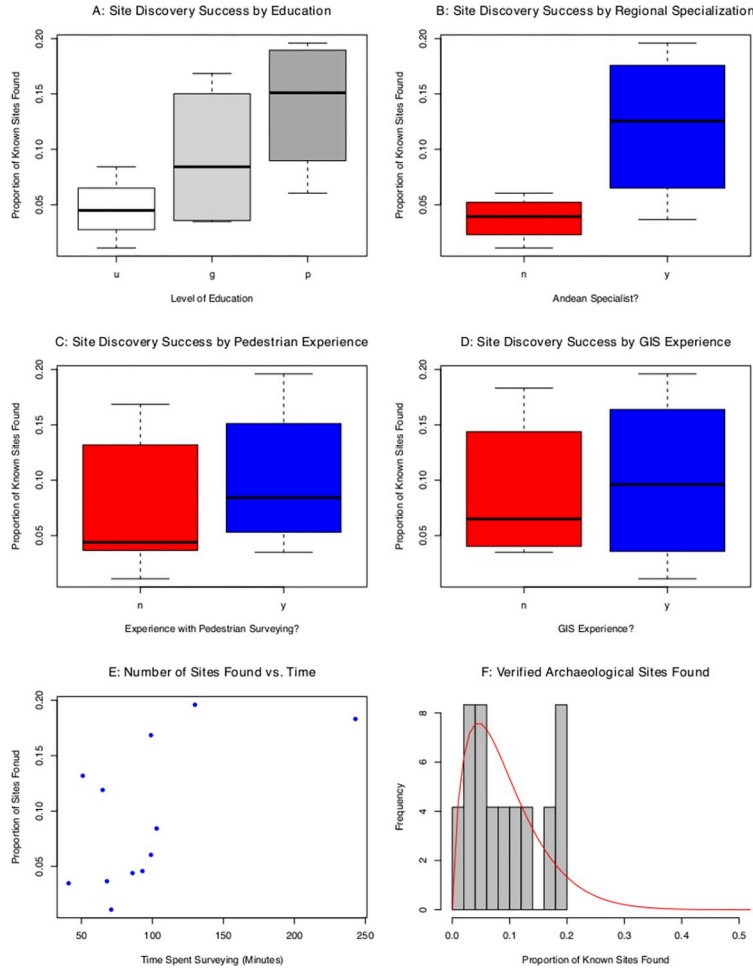

**Fig 2. Satellite survey site detection results.** Site discovery rate by A: Level of education (u = undergraduate student, g = graduate student, p = Ph.D.; *n* = 4 for each subset), B: Regional specialization (*n* = 8,4), C: Experience in pedestrian survey (*n* = 4,8), D: Experience with GIS (*n* = 7,5), E: Number of discovered sites as a function of survey time, F: Histogram of proportion of known sites found by the twelve participants. Red curve is the best-fit Beta distribution (KS D = 0.15, p = 0.90).

We find that formal archaeological training significantly affects site detection. With a mean discovery rate of 8%, graduate students produced a discovery rate roughly 70% higher than that of undergraduate students with a mean of 5%. Participants with a Ph.D. furthermore produced a discovery rate roughly 70% higher than that of graduate students, with mean discovery rates increasing from 8% to 13%, respectively. Increased time surveying produced a significant effect on discovery rates, with each additional minute spent surveying leading to a 1% increase in the likelihood of site discovery. Finally, experience with GIS was also found to have a strong effect on site discovery rate, producing an increase of approximately 106% from a mean of 5% to 10% between inexperienced and experienced GIS users (Fig 3).

## False positive rates

False positive rates ranged from 4–41% among participants. The best-fit beta model produces shape parameters of $\alpha$ = 1.4 and $\beta$ = 5.8 (Fig 3). A KS goodness-of-fit test shows the beta model to be a plausible model for the data (KS D = 0.16; *p* = 0.84). From the modeled Beta distribution, we derive a mean false-positive rate of 19% (2–52%).

**Table 3. Generalized linear mixed model results.**

| Model | Variable | Estimate | Standard Error | P-value |
|---|---|---|---|---|
| Site discovery | Education | -2.92 | 0.20 | $p < 0.01$ |
| | Regional Specialty | 5.88 | 0.88 | $p < 0.01$ |
| | GIS experience | 2.48 | 0.81 | $p < 0.01$ |
| | Pedestrian survey experience | -0.17 | 0.97 | 0.86 |
| | Time | 0.03 | 0.01 | $p < 0.01$ |
| False positive | Education | -1.04 | -0.24 | $p < 0.01$ |
| | Regional Specialty | 1.36 | 0.20 | $p < 0.01$ |
| | GIS experience | 0.61 | 0.34 | 0.07 |
| | Pedestrian survey experience | -0.61 | 0.20 | $p < 0.01$ |
| | Time | 0.01 | 0.00 | 0.03 |
| Taphonomy | Site Age | 0.00 | 0.00 | 0.31 |
| Site type | Chiribaya | -1.64 | 0.88 | 0.06 |
| | Estuquiña | -1.32 | 0.79 | 0.10 |
| | Huaracane | -0.80 | 0.45 | 0.08 |
| | Inka | -6.46 | 11.44 | 0.57 |
| | Omo | 1.93 | 0.90 | 0.03 |
| | Tumilaca | 1.65 | 0.57 | $p < 0.01$ |

Poisson general linear model analysis indicates that the following variables exert a significant ($p<0.1$) effect on false positive rates: educational experience, experience in Andean archaeology, and the amount of time spent surveying. Andean training had an effect on the frequency of false positives, with an increase of approximately 60%, from a mean of 10% to 16%. This surprising increase in error rate with greater experience likely reflects a sort of overconfidence among more experienced surveyors. An effect is also found between levels of educational experience. Graduate students found approximately 310% more false positives than undergraduate students, from a mean of 9% to 30%. However, there was no significant difference in the number of false positives between undergraduate students and Ph.Ds.

In sum, the GLMM shows that the most experienced surveyors–those with PhD degrees, regional experience, GIS experience, and pedestrian survey experience–produce mean false positive rates of 18% (8–23%), while the least experienced surveyors produce mean false positive rates of 15% (5–27%).

The site-age model did not identify a significant relationship between the latest occupation of an archaeological site and its probability of discovery ($p = 0.31$). In contrast, site type is observed to affect discovery rates. Omo and Tumilaca associated sites in the Middle Moquegua Valley were discovered at a higher frequency than the Chiribaya, Chen Chen, Estuquiña, Huaracane, and Inka sites ($p < 0.01$).

## Discussion

This study began by considering the efficacy of unstructured satellite survey in archaeological site detection. To evaluate this question, we investigated the quantitative utility of freely available satellite imagery for identifying archaeological sites in southern coastal Peru. Twelve study participants with varying degrees of experience performed satellite surveys in regions where archaeologists previously conducted high coverage pedestrian survey. Participants found an average of 9.3% (1–18%) of archaeological sites identified by ground based pedestrian survey. Our model further shows that highly trained individuals find 20% (17–22%) of archaeological sites in a given region, with the least trained individuals finding only 2% (1–

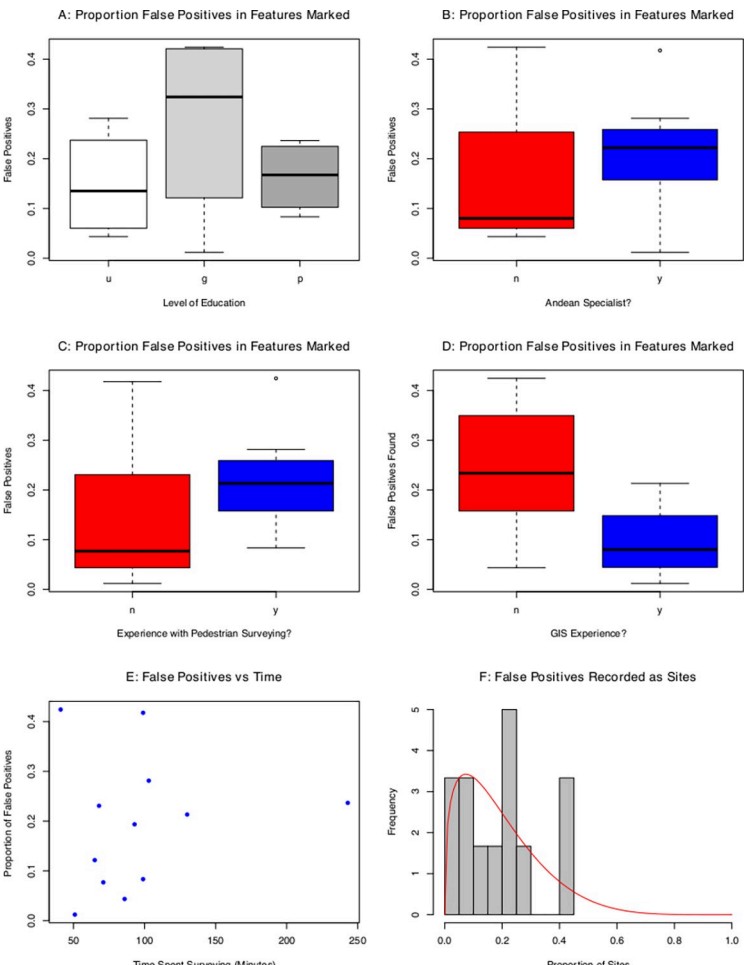

**Fig 3. False positive detection results.** False positive rate by A: Level of education (u = undergraduate student, g = graduate student, p = Ph.D.; $n$ = 4 for each subset), B: Regional specialization ($n$ = 8,4), C: Experience in pedestrian survey ($n$ = 4,8), D: Experience with GIS ($n$ = 7,5), E: Frequency of false positive features identified as a function of survey time, F: Histogram of proportion of false positives found by the twelve participants. Red curve is the best-fit Beta distribution (KS D = 0.17, p = 0.84).

3%) of sites. Although these results can be taken as strong evidence for a relationship between experience and site detection rates, the specific magnitude of effect should be treated cautiously given the relatively small participant pool and current limits in statistical theory. Nonetheless, the data are clear in showing that site-detection rates approaching 20% are readily attainable and that, unsurprisingly, the individuals with more training are more likely to approach such levels of success.

There are advantages and disadvantages to using an unstructured survey approach, rather than a more traditional structured one. First, unstructured searches require minimal training allowing researchers to take advantage of crowd sourcing approaches. Second, the lack of structure minimizes time spent strategizing the search plan and maintaining search structure. Third, unstructured surveys are likely to be more efficient, as surveyors gain information about hot spots as they go and can prioritize their approaches towards high density regions. Last, and potentially most importantly, unstructured searches are more engaging for participants who enjoy the organic discovery and exploration process.

Structured surveys, in contrast, can be tedious causing surveyors to lose focus and fail to identify sites in their field of view. This is not to suggest that unstructured surveys are superior to structured surveys. One major disadvantage of unstructured survey is the well-known risk of bias in areas that are preconceived to be productive while limiting search in other areas that may actually be archaeologically interesting. A second disadvantage is that unstructured surveys can result in search redundancy and thus inefficiencies in the use of time. Structured surveys thus yield more easily interpreted results [29]. Nonetheless, we acknowledge some of the advantages of unstructured survey along with the reality that it is a common exercise among practicing archaeologists.

Our analysis confirms some, but not all of our predictions regarding the effects of experience on site detection. The level of education, experience in Andean archaeology, experience working with GIS, and time spent surveying all impacted the number of verified archaeological sites found through satellite survey as anticipated. Not surprisingly, these data suggest that the more experience a researcher has working in the region of interest and working with remote sensing technology and GIS, the more likely they are to find archaeological sites through remote survey. Whether or not a researcher has spent time in the field itself participating in pedestrian survey was not found to have a significant effect on site detection in satellite imagery. More interestingly, we were able to quantify the degree to which different types and levels of training affect survey results, which theoretically allows analysts to calibrate satellite survey results based on the experience level of a surveyor.

Furthermore, we observed that site type, but not age, significantly affects the probability of site detection. One reason that site type may affect detectability is simply that certain culturally specific architectural elements are more visible in satellite imagery. Previous studies of remote survey note that archaeological sites located at high elevations and consisting of large geometric shapes—such as architectural structures—are much more likely to be detected than sites which lack these characteristics [54, 55]. A subtler reason may be related to the looting of archaeological sites. Looters tend to target certain archaeological sites associated with cultures that produced polychrome ceramics, colorful textiles, precious metals, and other goods that can be sold for profit on black markets [56]. The presence of 'looter pits' may actually change the visibility of archaeological sites in satellite imagery, making sites that would otherwise be invisible to satellite survey detectable.

Surveying a region through satellite imagery before entering the field, and then applying the correction-factors deduced from this analytical approach, may allow archaeologists to estimate the frequency of archaeological sites. For example, suppose a satellite survey of a previously unsurveyed region in coastal south-central Andes detected 100 sites. Assuming a satellite surveyor of unknown archaeological experience, our basic model would predict that anywhere between 555 to 10,000 sites (100/0.18 and 100/0.01, respectively) could be expected on the ground. Of course, this large estimate range is not particularly helpful. However, if we know something about the training of the satellite surveyor, we can derive tighter estimates. For example, if the surveyor were highly trained, then the model would estimate that 454–588 sites could be expected on the ground. Similarly, if the surveyor were minimally trained, the model would predict 3425–7388 sites on the ground. The wide range of possibilities in our uninformed model demonstrates that failure to account for experience strongly limits the predictive capacity of our model. The results show that archaeological experience exerts a significant effect on site discovery rates and accuracy. However, the reported extent of the effects should be interpreted with caution as should any attempt to use these models to predict site densities given surveyor experience. Additional research with increased participation could ultimately allow for more reliable parameter estimates.

Additional precision may be added to site frequency estimates with further research on the effects of independent survey variables. For example, it would be useful to further discriminate among other site types such as those with and without architectural features. Other factors to be considered in future research might include surveyor visual acuity and the specifications of computer equipment.

To the extent that our quantification of site discovery may be useful, these specific numerical results would only apply to southern coastal Peru and should be extended to similar contexts with caution. Due to the lack of tree and ground cover, archaeological sites in this region are not occluded in satellite imagery, making it an ideal region for this approach to site discovery [57]. Moreover, the scale and complexity of Andean architecture renders these sites exceptionally visible. In cases where the socio-ecological conditions appreciably differ from those of south-coast Peru, the method deployed here could be used to develop site-detection parameter estimates specific to the region of interest. Application of our method to other study regions may also have to account for other variables such as seasonal effects (e.g. vegetation change), which are virtually irrelevant in the Atacama and surrounding regions.

The use of our models as a quantitative guide for satellite survey is further restricted by the challenges of our relatively small sample size. Our participant population entirely consists of individuals pursuing degrees in anthropological archaeology or have dedicated their careers to archaeological research. We believe it is unlikely, for example, that an undergraduate student picked at random from an entire university population would perform as well as the undergraduate students who elected to participate in this study. However, this study and the consideration of its results are still of utility for archaeologists interested in performing or crowdsourcing satellite survey. Our results not only offer a tool for generating rough estimates archaeological site frequencies, but may also provide a baseline for evaluating the efficacy of machine learning approaches to remote survey. By establishing the capabilities of human satellite surveyors using freely available imagery under ideal conditions, we provide an important point of comparison for machine learning results–especially those conducted in south central Peru and other desert regions [58]. Such comparisons should furthermore permit assessment of the specific contexts in which machine learning succeeds or fails relative to small groups of experts.

One of the primary issues with manual remote survey—regardless of the type of imagery used—is inter-coder reliability [55, 59]. Sadr et al. (2016) find that different coders record archaeological features differently, regardless of training, boots-on-the-ground survey experience, or experience. However, this study was limited by their small sample size of only two individuals. The research project presented here helps to resolve the problem of inter-coder variability by demonstrating that coders different educational experience levels can have broadly similar behavior when recording archaeological sites through remote survey.

In sum, the efficacy of unstructured satellite survey may vary by experience and expertise, but even under optimal conditions (advanced education, experience in the region and with both GIS and pedestrian survey) satellite survey using freely available imagery only identifies ~20% of sites. Toward maximizing these results, regional expertise and knowledge of local history and geography remain some of the most important tools in the archaeologist's toolkit. In determining rates of error in satellite survey, archaeologists can intelligently generate rough estimates of site densities for planning field efforts, both of which are central tasks of archaeological research.

## Supporting information

**S1 File. Project R code.** All R code used in the analysis of raw data.
(DOCX)

**S2 File. Survey area extent data.**
(XLSX)

**S3 File. Known archaeological sites location data.**
(XLSX)

**S4 File. Participant result raw data.**
(XLSX)

## Acknowledgments

We thank all of the study participants for volunteering their time and energy. We also thank the Forager Complexity Lab members and the Davis Archaeology Working Group for feedback on the research design and results interpretation, as well as Bruce Owen for sharing pedestrian survey data. Finally, special thanks go to Sarah Baitzel and Anna Fancher Whittemore for their insight and feedback regarding early manuscript drafts.

## Author Contributions

**Conceptualization:** Thomas J. Snyder, Randall Haas.

**Data curation:** Thomas J. Snyder.

**Formal analysis:** Thomas J. Snyder, Randall Haas.

**Investigation:** Thomas J. Snyder.

**Methodology:** Thomas J. Snyder, Randall Haas.

**Writing – original draft:** Thomas J. Snyder.

**Writing – review & editing:** Randall Haas.

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
