## [Decision Letter · Decision Letter 0]

4 May 2023

PONE-D-23-03064Unstructured Satellite Survey Detects up to 20% of Archaeological Sites in Coastal Valleys of Southern PeruPLOS ONE

Dear Dr. Snyder,

Thank you for submitting your manuscript to PLOS ONE. After careful consideration, we feel that it has merit but does not fully meet PLOS ONE’s publication criteria as it currently stands. Therefore, we invite you to submit a revised version of the manuscript that addresses the points raised during the review process.

During the review process, a concern was raised that required consultation with the editorial board. I do apologize on behalf of the journal office for the delay this caused.

In their review of your manuscript, Reviewer #1 expressed concerns regarding the ability of the data to support the paperʼs conclusions. I share these concerns and ask that, as you address the reviewers' comments, you attend in particular to the limitations of the study’s very small sample(s) and the representativeness of the study participants, and that you consider reframing the paper as suggested by Reviewer #1. Additionally, Reviewer #2’s request for more information regarding properties of the satellite images used in the study is directly relevant to the study’s validity and should be addressed in the revisions.

We look forward to receiving your revised manuscript.

Kind regards,

Raven Garvey, Ph.D.

Academic Editor

PLOS ONE

Journal Requirements:

2. Please ensure that you have specified (1) whether consent was informed and (2) what type you obtained (for instance, written or verbal, and if verbal, how it was documented and witnessed). If your study included minors, state whether you obtained consent from parents or guardians. If the need for consent was waived by the ethics committee, please include this information.

“No, this research and data collection required no funding.”

4. We note that Figures 1 and 2 in your submission contain map and satellite images which may be copyrighted. All PLOS content is published under the Creative Commons Attribution License (CC BY 4.0), which means that the manuscript, images, and Supporting Information files will be freely available online, and any third party is permitted to access, download, copy, distribute, and use these materials in any way, even commercially, with proper attribution. For these reasons, we cannot publish previously copyrighted maps or satellite images created using proprietary data, such as Google software (Google Maps, Street View, and Earth). For more information, see our copyright guidelines: http://journals.plos.org/plosone/s/licenses-and-copyright.

 a.          You may seek permission from the original copyright holder of Figures 1 and 2 to publish the content specifically under the CC BY 4.0 license. 

Reviewers' comments:

Reviewer's Responses to Questions

**Comments to the Author**

1. Is the manuscript technically sound, and do the data support the conclusions?

Reviewer #1: Partly

Reviewer #2: Partly

2. Has the statistical analysis been performed appropriately and rigorously? 

Reviewer #1: Yes

Reviewer #2: Yes

3. Have the authors made all data underlying the findings in their manuscript fully available?

Reviewer #1: Yes

Reviewer #2: Yes

4. Is the manuscript presented in an intelligible fashion and written in standard English?

Reviewer #1: Yes

Reviewer #2: Yes

5. Review Comments to the Author

Reviewer #1: I should begin by noting that I have reviewed this manuscript twice before for another journal. I continue to see the issue of assessing success rates of unstructured satellite survey as of fundamental interest to many archaeologists, and the findings are more clearly presented than they initially were. However, in my read the paper still relies too much on quantitative analysis of a small dataset that is asked to support quite a lot. The quantitative analyses are undermined by the small sample size, making it a mistake in my opinion to focus on them. Rather, I would suggest minimizing the attempt to squeeze additional interpretations out of these data, and focusing instead on the central finding that detection rates are quite low.

In fact I think even rough approximations of detection and false positive rates from unstructured satellite survey are quite valuable. Even those generalizations, however, are built on a sample of only twelve. Focusing on the categorical separation of those twelve into three types with only four samples each puts even the most rigorous quantitative analysis on shaky ground. Do the groups of participants constitute random and/or representative samples of undergrads/grads/PhDs? With respect to which particular aspects of those populations? Are the differences between the groups large enough – given the small sample sizes – to constitute compelling evidence about the populations in question? The variances in sites found and false positives in each group suggest that these small samples are *very* heterogenous and thus likely to not characterize the populations (of undergrads/grads/PhDs) very well.

I noted something similar in a previous review and I think this comment (lightly edited here) remains applicable to the current version:

To my mind, the overall results have interesting and useful things to say about likely success rates of imagery-based survey (even under what are, I would judge, effectively pretty optimal conditions on the S. Coast of Peru). That central message, to me, is a much more important (and robust) result than the attempts to identify characteristics of participants that affect their success rates. With respect to the latter, sample sizes are quite small (4 each of undergrad, grad, and PhD students), and no attempt is made to argue that the samples are in any way representative (nor do I imagine that such an argument would be possible). To me this suggests that some re-emphasis make for a more compelling paper, with a central message something like, “results may vary by experience and expertise, but even under optimal conditions (advanced education, experience in the region and with both GIS and pedestrian survey) satellite survey only identifies ~20% of sites”. That is, the variables considered apparently can matter, but even in the best-case scenario identification rates are quite low.

A few specific comments:

The last two paragraphs of the intro read more like a grant proposal than an article, to me. I’d suggest that here the focus should be on what the *findings* are, rather than what the hypotheses are.

There are several instances of “Error! Bookmark not defined.” codes appearing in the text (p4, 6, 13).

p5

“Further hampering all satellite survey approaches is the fact that features of archaeological sites that may be visible via satellite imagery may be obscured by modern architecture and therefore invisible to satellite photographs without field observations, excavation, or ground penetrating radar (33).”

I find this confusingly written, and would strike out the clause “that may be visible via satellite imagery”. I’d also suggest replacing “modern architecture” with the broader “modern development”.

p7

“While there are no pre-existing models to suggest the proportion of archaeological sites that can be found through unstructured satellite survey, clearly the proportion must lie between 0–100%. We further expect the proportion to be on the low end of this range”

I would strike out “clearly the proportion must lie between 0–100%”, since it’s stating the obvious. The following sentence could be simplified to just, “We expect the proportion to be low”.

p8

“Third, the lack of dense surface vegetation is ideal for remote detection by satellite imagery.”

Should be reworded to something like, “lack of dense surface vegetation provides ideal conditions for remote detection”.

p9 Section on “Satellite Survey”

This whole section should not be in the present tense. i.e., “were recruited to perform” rather than, “are recruited to perform”.

p16

“Our binomial GLMM results demonstrate that the following variables significantly (p < 0.1) affect site detection rates: educational experience, experience in Andean archaeology, experience working with GIS, and time spent surveying”

These variables are not independent of one another: experience in Andean archaeology and working with GIS are likely to be correlated with educational experience.

p19

I’m not sure I follow the distinction between site age and site type that’s used here. The site types listed are from time periods that are (to an extent) distinct.

p20, Ln 375

“site discovery rates are likely to be much higher than in unstructured surveys”

Should this read “structured surveys”?

Table 2

Row 1, ‘Sites Found’ column reads “25 (4%)”, but should read “25 (.04)” to match the other rows.

Figs. 3 and 4:

Some other color than red should be used for the ‘p’ boxplot in (A), to avoid the impression of comparable coding with B, C, and D. In fact, it’s not clear that color is adding anything to these boxplots at all. All of them should also specify the samples sizes.

For (E) in both figures, either the y-axis tick labels or the y-axis is mislabeled. For (F), these appear to be showing frequency rather than density and the y-axis should reflect that.

Reviewer #2: An interesting and relevant study that I hope to see published. However, the authors should address the following aspects in their revision:

Image data: No information about the image data is provided, even though it is a crucial factor in the analysis. While commercial portals such as the ones used provide no detailed data, they do provide some basic data, e.g. in Google Earth/Maps the satellite/sensor used and the year of acquisition. This gives some indication of image properties and quality, which may vary greatly across the study areas and thus affect the results. The second parameter is important as it defines the time passed between the archaeological survey and the time the images were acquired, during which the situation on the ground might have changed.

Archaeological data: The site datasets are generally well chosen but do not represent the full archaeological record of the study areas, which the authors fails to discuss. The survey projects probably missed some sites while others were lost since. Furthermore, the sites probably vary greatly in type, size, preservation etc., but this is not discussed either even though it is another important parameter for the task in question.

Related research: There has been more research undertaken on the general topic of this article than the authors indicate, e.g. on coder reliability (https://doi.org/10.1002/arp.1515) and other factors affecting site detection rates (https://doi.org/10.1017/aap.2017.13;
http://dx.doi.org/10.1016/j.jas.2013.07.002). These studies should be taken into account in the introduction and/or discussion. Furthermore, there are relevant recent projects in Peru (https://iopscience.iop.org/article/10.1088/2051-672X/ac9492) that the authors should cite for context. In addition, the following studies seem relevant for some points that the authors raise in the discussion: https://doi.org/10.3390/geosciences8080272 on the varying reliability of the mapping results of experts, which often serve as benchmark; https://doi.org/10.3390/rs11070794 on integrating automation and crowdsourcing; and https://hdl.handle.net/1887/3256824 (especially ch. 7) on the question to which level of human expertise the results of automated mapping of archaeological sites should be compared. Considering these studies will enrich and broaden the discussion of the research described here.

Finally, a number of minor editorial issues that should be addressed:

50: in the Yucatan => in Yucatan

82: the process site discovery => the process of site discovery

99: to search successfully search => to successfully search

219: not reported for the Goldstein 2005, but => not reported for the Goldstein (2005) survey, but

Table 2, participant 1: (4%) => (0.04)

424: factors might be considered in future research might include => factors to be considered in future research might include

434: other variables such as seasonal effects such as vegetation changes => other variables such as seasonal effects (e.g., vegetation changes)

475: Journal of archaeological science => Journal of Archaeological Science

488: International journal of applied earth observation and geoinformation => International Journal of Applied Earth Observation and Geoinformation

522: Incomplete reference

558: Journal of archaeological Science => Journal of Archaeological Science

580: In: th Annual Meeting => add number

Fig. 3E: Number of sites found => Proportion of sites found

Fig. 4E: Number of false positives => Proportion of false positives

Supporting information 2, column B heading: Eating => Easting

Supporting information 4, column C heading: Participant => Northing

6. PLOS authors have the option to publish the peer review history of their article (what does this mean?). If published, this will include your full peer review and any attached files.

Reviewer #1: No

Reviewer #2: No

---

## [Author Response · Author response to Decision Letter 0]

28 Jun 2023

Detailed responses to reviewer and editor comments are located in the uploaded 'Cover Letter' and 'Response to Reviewer Comments' documents.

---

## [Editor Report · Decision Letter 1]

10 Jul 2023

PONE-D-23-03064R1Unstructured Satellite Survey Detects up to 20% of Archaeological Sites in Coastal Valleys of Southern PeruPLOS ONE

Dear Dr. Snyder,

I have assessed the revision and note that Reviewer #1’s previously-raised, valid concerns regarding the ability of the data to support the paper’s conclusions is insufficiently addressed. The journal’s criteria for publication (https://journals.plos.org/plosone/s/criteria-for-publication) state that “sample sizes must be large enough to produce robust results.” Reviewer #1 rightly notes that, especially when parsed by education level and other measures of experience, your data do not appear to meet this requirement. While you have included a paragraph in the discussion acknowledging the small number of study participants, the manuscript remains largely unchanged, a decision that is not adequately justified in your rebuttal letter and does not address the issue of the results’ robusticity. Ordinarily, this would constitute grounds for rejection, but I have chosen to provide you with another opportunity to revise the manuscript before issuing a final decision. If you choose to undertake a second revision, I strongly urge you to consider Reviewer #1’s concerns more carefully and to reframe the paper in terms of the general result (low success rates of imagery-based survey), limiting and fully qualifying any discussion of participants’ performance based on experience and expertise. If these concerns are not fully addressed in the revision, I may reject the manuscript without further review.

We look forward to receiving your revised manuscript.

Kind regards,

Raven Garvey, Ph.D.

Academic Editor

PLOS ONE

---

## [Author Response · Author response to Decision Letter 1]

31 Aug 2023

Dear Dr. Garvey:

We appreciate the additional clarification you have provided and, again, the opportunity to address the shortcomings of our analysis. The concerns of the reviewer have become clearer to us. We now understand the statistical limitations that raise concerns about our effect-size claims. We agree that current theoretical limitations coupled with our small participant pool indeed limits our ability to specify parameter values for the identified relationships. We have therefore revised our text to reflect this limitation and have also clarified some of our analytical choices with particular attention to those that pertain to statistical power. These edits include the following:

edited line 291 as follows: "...we use generalized linear mixture modelling (GLMM) using the Satterthwaite approximation applied to restricted maximum likelihood fitted models as implemented with the lme4 (Bates et al 2015) and lmerTest packages (Kuznetsova et al. 2017) in R statistical computing environment (R Core Team)." (https://www.jstatsoft.org/article/view/v067i01, https://www.jstatsoft.org/article/view/v082i13)

Line 311, added "These data provide the basis for estimating site-detection rates, false-positive rates, and the effects of surveyor experience on those rates. However, the relatively small participant pool raises concern about the ability to accurately characterize effect sizes. A recent simulation study shows that although GLMM is relatively robust in terms of detecting effects (Luke 2017), the effect of sample size on accurate quantification of effect size remains poorly understood. We therefore proceed with confidence in interpreting the model's ability to detect relationships but with caution in terms of the model's ability to quantify effect sizes." (https://link.springer.com/article/10.3758/s13428-016-0809-y)

Line 381, added, "Although these results can be taken as strong evidence for a relationship between experience and site detection rates, the specific magnitude of effect should be treated cautiously given the relatively small participant pool and current limits in statistical theory. Nonetheless, the data are clear in showing that site-detection rates approaching 20% are readily attainable and that, unsurprisingly, the individuals with more training are more likely to approach such levels of success."

Line 433, edited to "The results show that archaeological experience exerts a significant effect on site discovery rates and accuracy. However, the reported extent of the effects should be interpreted with caution as should any attempt to use these models to predict site densities given surveyor experience. Additional research with increased participation could ultimately allow for more reliable parameter estimates."

Line 440, edit to "To the extent that our quantification of site discovery may be useful, these specific numerical results would only apply to..."

We also corrected an error in reported values in table 3. The changes do not affect overall interpretation. We respond to other comments in our response to reviewers and cover letter. We hope that these corrections address the reviewer's concerns, and we look forward to your response. 

Best,

Thomas Snyder and Randy Haas

---

## [Editor Report · Decision Letter 2]

18 Sep 2023

Unstructured Satellite Survey Detects up to 20% of Archaeological Sites in Coastal Valleys of Southern Peru

PONE-D-23-03064R2

Dear Dr. Snyder,

We’re pleased to inform you that your manuscript has been judged scientifically suitable for publication and will be formally accepted for publication once it meets all outstanding technical requirements.

Kind regards,

Raven Garvey, Ph.D.

Academic Editor

PLOS ONE
---

## [Editor Report · Acceptance letter]

25 Sep 2023

PONE-D-23-03064R2 

Unstructured satellite survey detects up to 20% of archaeological sites in coastal valleys of southern Peru 

Dear Dr. Snyder:

I'm pleased to inform you that your manuscript has been deemed suitable for publication in PLOS ONE. Congratulations! Your manuscript is now with our production department. 

Kind regards, 

on behalf of

Dr Raven Garvey 

Academic Editor

PLOS ONE